# Consensus Sequences for *Gag* and *Pol* Introduced into HIV-1 Clade B Laboratory Strains Differentially Influence the Impact of Point Mutations Associated with Immune Escape and with Drug Resistance on Viral Replicative Capacity

**DOI:** 10.3390/v17060842

**Published:** 2025-06-12

**Authors:** Sven Breitschwerdt, Benedikt Grandel, Benedikt Asbach, Franziska Winter, Todd Allen, Ralf Wagner, Bernd Salzberger, Arne Schneidewind

**Affiliations:** 1Department of Internal Medicine I, University Hospital Regensburg, 93053 Regensburg, Germany; 2Department of Internal Medicine I, University Hospital Bonn, 53127 Bonn, Germany; 3Institute of Medical Microbiology and Hygiene, University Regensburg, 93053 Regensburg, Germany; benedikt.asbach@klinik.uni-regensburg.de (B.A.);; 4The Ragon Institute of MGH, MIT and Harvard, Boston, MA 02139, USA; 5Institute of Clinical Microbiology and Hygiene, University Hospital Regensburg, 93053 Regensburg, Germany; 6Department of Hospital Hygiene and Infectious Diseases, University Hospital Regensburg, 93053 Regensburg, Germany

**Keywords:** HIV, viral backbone, consensus sequence, immune escape, replication capacity, drug resistance

## Abstract

Viral evasion from effective human immunodeficiency virus type 1 (HIV-1)-specific CD8+ T-cell responses and from antiretroviral therapy through viral sequence variation is frequently accompanied by a loss in viral fitness. The impact of sequence variations on replication capacity in vitro was mostly studied by introducing single mutations into a specific clonal strain such as NL4-3. How the specific viral backbone itself impacts replicative fitness remains elusive. To test for a potential effect of the viral backbone, we constructed HIV-1 clade B clones with consensus sequences for *gag* and/or *pol* and evaluated the infectivity of viral variants harboring well-defined cytotoxic T-lymphocyte (CTL) escape mutations or drug resistance mutations within this backbone or the clonal NL4-3 strain. Viral variants with consensus sequences were replication-competent in vitro, although at lower rates than the NL4-3 virus. Introduction of the dominant CTL escape mutation R_264_K into the newly constructed viruses or into NL4-3 led to a dramatic reduction in infection rates. In contrast to the NL4-3 backbone, the combination of R_264_K with its compensatory mutation S_173_A on the consensus backbone led to higher infection rates as compared to the same virus in the absence of R_264_K and S_173_A. Furthermore, 2 out of 10 drug resistance mutations in *pol* led to opposing effects, with an increase in infection rates on the consensus *gag/pol* backbone and a reduction on NL4-3. Therefore, the effect of the respective viral backbone on infectivity observed in vitro might constitute an additional factor to explain differential kinetics of mutational evasion from immune and pharmaceutical pressure.

## 1. Introduction

The interplay between human immunodeficiency virus type 1 (HIV-1) and the human immune response remains incompletely understood. There is strong evidence that CD8+ cytotoxic T-lymphocyte (CTL) responses established during early infection enable the host to transiently control viral replication [1,2,3,4]. Subsequently, immune pressure by these responses leads to the selection of viral variants with sequence alterations in targeted epitope regions, allowing the virus to escape CTL recognition and to achieve higher levels of viral replication [5,6]. It is also known that T cell responses linked to specific human leukocyte antigen (HLA) class I alleles are associated with slower disease progression. Examples are HLA-B*27 and HLA-B*57, which seem effective especially in the early stages of HIV infection [7,8,9,10,11] and are over-represented among long-term non-progressors and elite controllers [12,13,14]. Specific viral epitopes targeted by HLA-B*27- and HLA-B*57-restricted CD8+ T cell responses are highly conserved, a prime example being the HLA-B*27-restricted KK10 epitope (K_263_–K_272_) in *gag*. When escape mutations leading to diminished CTL recognition arise in these conserved epitopes, they are typically associated with a loss in viral replication capacity [15,16,17,18,19].

A major aim of combination antiretroviral therapy (cART) in people living with HIV (PLHIV) is a complete suppression of viral replication in order to enable durable reconstitution of an effective immune system and thereby reduce HIV-infection-associated morbidity and mortality. Selection of drug resistance mutations (DRMs) in *pol* leads to reduced drug susceptibility and detectable viral replication despite cART [20]. Persistent viral replication in PLHIV does not necessarily lead to a continuous loss of CD4+ cells and to clinical failure [21,22,23,24,25]. This protection from disease progression could be the consequence of an overall reduction in viral fitness through the combined effect of DRMs with the remaining antiviral effect of cART. Accordingly, the in vitro replication capacity of viral variants harboring DRMs has been shown to be generally lower than that of variants without such mutations [23,26,27,28,29,30,31,32]. Moreover, in vivo, the negative impact of DRMs on viral replication capacity is also demonstrated by the rapid resurgence of viral variants without the previously selected DRM after discontinuation of cART [23,33,34,35,36,37] and by mutational reversions after transmission to a new host [38,39,40]. In addition, the negative impact of DRMs on replicative fitness can be overcome by the addition of secondary mutations capable of restoring infection rates [41,42], potentially followed by more rapid disease progression [43].

In longitudinal and cross-sectional sequencing studies, the kinetics for selection of CTL escape mutations appear to vary substantially for different epitopes, and an elicited immune response does not always result in CTL escape variants [8,16,43,44,45,46]. Similarly, the time to resistance acquisition under the selective pressure of cART in clinical cohort studies varies significantly among PLHIV. In PLHIV initiating treatment, acquisition of DRMs is usually observed within days to a few weeks on monotherapy [47,48,49,50] and within weeks to few months [51,52] on dual therapy. In contrast, when cART comprises three antiretroviral substances, DRMs remain a rarity even after years of treatment [20,53]. The probability for the selection of specific DRMs in vivo [54,55] and in vitro [56] also varies significantly dependent on the HIV-1 subtype, suggesting differences in the genetic barrier for the selection of DRMs.

The impact of point mutations associated with CTL escape is typically tested by inserting them into specific clonal HIV-1 strains such as NL4-3 [57] or HXB2 [58]. These primary clones isolated from the quasispecies of a person infected with HIV contain random mutations and mutations originating from in vivo selection pressure in the host, which might affect replication in vitro, thus making it difficult to obtain generalizable results. We therefore used HIV-1 clade B consensus sequences from the Los Alamos National Laboratories (LANL) HIV database [59]—derived by alignment of available primary sequences—to construct proviral variants of the laboratory strain NL4-3 harboring consensus sequences of the *gag*, *pol*, and *gag/pol* genes. We hypothesized that using a proviral consensus backbone to study the effects of individual point mutations should better pinpoint the impact on replication capacity than by utilizing a randomly selected clone. Thus, in this study, we compared infection rates as a surrogate parameter for replicative capacity in consensus backbones versus the NL4-3 backbone. We tested the consequences of inserting different mutations associated with CTL escape in *gag* and with drug resistance in *pol*. Our data confirm a substantial influence of the respective viral backbone on replication capacity in vitro. The backbones’ differential impact on replication capacity might represent a cofactor determining the varying kinetics for selection of drug resistance mutations or CTL escape mutations in vivo.

## 2. Materials and Methods

### 2.1. Consensus Sequence

The templates of the consensus sequences for gag and pol were derived by aligning sequences from all primary HIV-1 Clade B isolates deposited in the database of the Los Alamos National Laboratories (http://www.hiv.lanl.gov/content/sequence/HIV/mainpage.html) accessed on 8 March 2007. Alignment and generation of a consensus sequence was facilitated by using the Consensus Maker tool by the Los Alamos National Laboratories. The consensus sequences of gag and pol differed from NL4-3 at 13 and 11 amino acid positions, respectively.

### 2.2. Variant NL4-3 and Consensus Constructs

In order to construct the consensus variants, as well as inserting drug resistance mutations, HIV-1 strain NL4-3 was used as template. The newly generated consensus variants were later used as viral backbones for the immune escape and drug resistance mutations. To generate consensus sequences of *gag* and *pol*, a *SacI-SbfI* or *SbfI-SalI* fragment (nt263-947nt or nt948-nt1927) was excised from proviral pNL4-3 and cloned into the vector pUC19. The 13 mutations for *gag* and the 11 mutations for *pol* required to convert the NL4-3 sequence into the respective consensus variants were inserted by site-directed mutagenesis. Individual intermediates, as well as the final consensus variants, were verified by sequence analysis. For mutagenesis, the GeneTailor, the GeneArt site-directed mutagenesis system (Invitrogen), or the QuickChange II site-directed mutagenesis kit (Agilent, Santa Clara, CA, USA) were used as previously described [16]. Mutagenesis was performed by using oligonucleotide primers (Appendix A). The complete HIV-1 coding region of the variant proviruses was sequenced by using published primers [6,60]. *Escherichia coli* One Shot Stbl3 cells (Invitrogen) were used to propagate full-length proviral plasmids, and stocks were prepared by using a QIAprep Spin Miniprep kit or HiSpeed Plasmid Midiprep kit (Qiagen, Düsseldorf, Germany).

### 2.3. Cells

HEK293T cells were cultured at 37 °C in a 5% CO_2_ incubator in Dulbecco’s Modified Eagle Medium (DMEM; PAA) supplemented with 10% fetal calf serum (Gibco), penicillin (100 IU/mL), and streptomycin (100 µg/mL) (Sigma). CEM-GXR25 [61] cells were cultured in R10^+^ medium (RPMI 1640 medium [Sigma] supplemented with 10% fetal calf serum (FCS, Gibco, Waltham, MA02451, USA), 2 mM L-glutamine, penicillin [100 IU/mL], and streptomycin [100 µg/mL] (Sigma, Saint Louis, MO 63103, USA)) at 37 °C and 5% CO_2_.

### 2.4. Viral Stocks

Viral stocks were generated by transfection of HEK293T cells with 4 µg of plasmid DNA. Cells were cultivated in antibiotic-free DMEM 24 h before transfection. Transfection was performed by using Lipofectamine 2000 (Invitrogen, Grand Island, NY 14072, USA) in OptiMEM (Invitrogen). Supernatants were harvested, cleared, and analyzed for p24 content by ELISA 48 h after transfection. Frozen aliquots were stored at −80 °C.

### 2.5. ELISA

Capsid concentration of the viral stocks was quantified by p24 enzyme-linked immunosorbent assay (ELISA) by using the Alliance HIV-1 p24 ELISA kit (Perkin-Elmer, Shelton, CT 06484-4794, USA). The ELISA was performed according to the user manual with the alteration of adding 1% FCS to the PBS used for sample dilutions.

### 2.6. Viral Infectivity Assays

One million CEM-GXR25 cells were inoculated with 250 ng p24 from the respective viral stocks, and the volume was adjusted to 1 mL by adding R10^+^ medium. Wells were replenished 6 h post-infection to a volume of 2 mL with R10^+^. After 48 h, 1 mL of each culture was harvested.

### 2.7. Flow Cytometry

CEM-GXR25 cells, expressing a plasmid encoding a humanized enhanced green fluorescent protein (GFP) under the control of an HIV-1 long terminal repeat [62], were fixed in phosphate-buffered saline containing 4% paraformaldehyde, and infectivity was determined by detecting the proportion of green fluorescent protein (GFP)-expressing cells by using flow cytometry (FACS Canto II flow cytometer and FACS Diva 7.0 software (Becton Dickinson, Franklin Lakes, NJ, USA, 07417-1880)). Background signal was defined as 0.01% of the uninfected negative controls, which was subtracted from the respective samples. A minimum of 200,000 cells were analyzed for each sample. The percentages of GFP^+^ cells were normalized to wild-type virus NL4-3 or the respective consensus viral variant.

### 2.8. Statistical Analysis

Data (percentages of infected cells in independent assays, normalized to mean value of percentage in wild-type NL4-3) were analyzed in a generalized linear model with viral backbones and single mutations as fixed factors (SPSS 28.0, SPSS Inc., Chicago, IL, USA). Post hoc comparisons were performed with Bonferroni corrections for multiple comparisons.

## 3. Results

### 3.1. HIV-1 Clones with Consensus Sequences in Gag and/or Pol Are Replication-Competent

We first tested the replicative fitness of three *gag* and/or *pol* consensus HIV-1 clones. In comparison to NL4-3, the HIV-1 consensus sequence differs in 13 amino acid positions in *gag* (Figure 1A) and in 11 positions in *pol* (Figure 1B). Primary viral isolates containing a similar set of consensus sequences have not been described from clinical samples. Initially, we wanted to assess whether a viral clone with a consensus sequence was replication-competent at all. We therefore generated three different viral clones on the NL4-3 backbone using site-directed mutagenesis. Two of the novel viral clones carried the consensus sequence for *gag* or *pol*, respectively, and a third contained the consensus sequences for both *gag* and *pol*.

All three viral clones with consensus sequences were replication-competent in vitro. Infection rates of viral particles with consensus *gag* reached a factor of 0.54, viral particles with consensus *pol* reached a factor of 0.46, and those with consensus *gag* and *pol* reached a factor of 0.40 of NL4-3 (Figure 2). Infection rates of all three consensus variants were significantly lower than that of NL4-3. Differences in replication capacity between the three consensus variants were not statistically significant (Appendix A. Thus, whereas the reduction in infectivity was most pronounced for the consensus *gag/pol* variant, there seems to be no pure additive effect on infectivity by combining consensus *gag* and *pol* sequences.

### 3.2. The Impact of Immune Escape Variants on Replication Capacity in Gag Is Influenced by the Underlying Backbone

Having demonstrated sufficient replication competence of viral variants with consensus sequences in *gag* and/or *pol*, we next investigated the differential impact of specific point mutations associated with CTL escape on viral replication in the consensus versus the NL4-3 virus backbones. We first focused on the impact of three point mutations (S_173_A, R_264_K, and L_268_M) associated with immune selection pressure against the HLA-B27-restricted CTL-epitope KK10 in *gag*. These mutations and their impact on viral fitness have been well defined previously by using NL4-3 as a virus backbone [16]. Mutations R_264_K and L_268_M within the KK10 epitope itself abrogate epitope binding to the HLA-B27 molecule, resulting in reduced T-cell recognition of the infected cell. In contrast, S_173_A is a secondary mutation [63] compensating for the severe functional replication defect introduced by the primary escape mutation R_264_K. L_268_M and S_173_A alone or in combination with each other had no significant effect on replication competence [8,43,64,65].

We generated viral constructs containing mutations S_173_A, R_264_K, and L_268_M individually, as well as double and triple mutants, on the newly generated consensus *gag* backbone and compared their replicative capacities in vitro with that of the *gag* consensus NL4-3 variant (Figure 3). The infection rates of the variant carrying the primary CTL escape mutation R_264_K (RK on Cons gag) was significantly reduced to 0.14 of the unmodified consensus *gag* construct, whereas the combination of R_264_K with L_268_M (RKLM on Cons gag) had less impact on replication, with a factor of 0.27 (Figure 3 and Appendix A). In contrast, compensatory mutation S_173_A led to increased infection rates beyond the Cons gag standard both in the absence and presence of R_264_K (1.18 for SA on Cons *gag*, 1.54 for SARKLM on Cons *gag*, and 1.33 SARK on Cons *gag*, respectively).

Taken together, the previously described negative consequences of different KK10 escape mutations on the NL4-3 backbone on infection rates [42] were also observed in the novel viral construct with the artificial consensus sequence for *gag*. In contrast to previous results on the NL4-3 backbone [42], the introduction of the compensatory mutation S_173_A into the consensus construct carrying the two primary escape mutations (SARKLM on cons gag) not only compensated for the replication defect of the primary escape mutation R_264_K but also resulted in greater replication capacity than that of the unmutated consensus *gag* backbone.

### 3.3. The Viral Backbone Differentially Influences the Impact of Drug Resistance Mutations on Replication Capacity

We next investigated the impact of specific drug resistance mutations on the replicative fitness of viral consensus variants. We constructed a panel of ten viral variants with clinically relevant drug resistance mutations in the original NL4-3 backbone or in the newly designed *gag* and *pol* consensus variants. Mutations K_65_R, M_184_V, and Q_151_M mediate resistance against multiple nucleoside reverse-transcriptase inhibitors (NRTIs) [66,67,68,69], whereas the other mutations mediate resistance against non-nucleoside reverse-transcriptase inhibitors (NNRTIs) (L_100_I, K_101_P, Y_181_C), protease inhibitors (D_30_N, L_76_V, I_84_V), or integrase inhibitors (N_155_H) [69,70,71,72,73,74,75].

Consistent with previously published data [76,77,78,79] for the NL4-3 backbone, most drug resistance mutations led to a reduction in infected cells in vitro at different levels. Only the variant with the L_100_I mutation replicated at higher levels than NL4-3 (factor 1.06), whereas the remaining nine drug resistance mutation variants led to reduced infection rates, levels between 0.91 (Y_181_C) and 0.23 (L_76_V) were recorded (Figure 4 and Appendix A).

Finally, we tested the impact of the different drug resistance mutations in the *gag/pol* consensus sequence backbone (Figure 5 and Appendix A). Similar to the experiments with the original NL4-3 backbone, the introduction of mutation L_100_I led to an elevation in infection rate as compared with the *gag/pol* consensus construct. In contrast to the respective experiments utilizing the NL4-3 backbone, two other drug resistance mutations, K_65_R and M_184_V, also led to higher infection rates than the *gag/pol* consensus virus (relative infection rates of factor 1.37, and 1.40, respectively). The remaining seven variants had similar reductions in their infection rates compared to the experiments with the NL4-3 backbone. Importantly, infection rates for all viral variants with consensus sequences were significantly lower compared with NL4-3 (Appendix A).

In summary, we demonstrate major differences in the impact of specific immune escape and drug resistance mutations on the replicative capacity of viral lab strain NL4-3 versus a viral backbone with artificial consensus sequence. Occasionally, specific mutations had a detrimental impact on infection rates in one viral backbone and an enhancing effect on infection rates in the other viral backbone.

## 4. Discussion

Viral escape from CTL responses and development of resistance to antiviral drugs are common among highly mutable viruses such as HIV [80,81]. Typically, viral escape from CTL selection pressure occurs through selection of viral variants with one or a few single amino acid exchanges in the targeted epitopes or its flanking regions, leading to diminished MHC presentation or TCR stimulation [64,82]. These sequence variations abolish or diminish recognition by CD8+ T cells but often come with a cost to the virus, i.e., a loss in viral fitness [16,83,84,85]. Similarly, viral variants that are less susceptible to antiviral drugs may also exhibit inferior replicative fitness. Sometimes, secondary mutations arise later that at least partially restore viral replication [41,44,86,87,88,89,90,91]. Such secondary, or compensatory, mutations can be several dozen amino acid positions away from the original mutation and, in the case of CTL escape mutations, are often outside the CTL epitope region. Considering the complex effects of combinations of multiple specific amino acid exchanges at distant locations, we hypothesized that the impact of CTL escape mutations and drug resistance mutations could be significantly different in distinct viral strains based on their overall genetic diversity and associated replicative capacity.

Analyses of the impact of primary and secondary mutations on replication capacity were usually carried out by introducing the respective amino acid changes into primary isolates such as NL4-3 [92,93,94,95,96,97] that were isolated from PLHIV. Typically, the sequence of these primary isolates has been shaped by host immune selection pressure. Thus, primary and secondary mutations that alter replicative fitness may already be present. To what degree these adaptations impact in vitro studies of additional mutations remains unclear. We therefore constructed proviral variants with consensus sequences for *gag*, *pol*, and *gag/pol*, with the aim to create a sequence backbone that is less biased by specific host adaptations and their impact on replication capacity. We hypothesized that this would allow us to gain additional insights into the effect of individual amino acid exchanges that might be reflective of a wider range of naturally occurring viruses than the results from a viral lab strain.

First, we demonstrated that all three viral variants containing consensus sequences in *gag* and/or *pol* were replication-competent, though they replicated at lower rates compared with the lab strain NL4-3. This difference in infection rates may be explained by the fact that NL4-3 was originally isolated from a patient’s quasispecies and may represent a virus with optimized replication capacity in vivo. In contrast, our newly constructed consensus strains do not occur in vivo but rather represent an artificial average of numerous individual B-clade clones. These different viral clones all carry footprints in their respective amino acid sequence that arose by certain selection pressures. Because of these sequence footprints, we conclude that either one or the combination of several mutations distinguishing the consensus strains from NL4-3 (13 mutations in *gag*, 11 in *pol*, and 24 in *gag/pol*) would functionally impair replication capacity to a certain degree. Nevertheless, the viral variants with consensus backbones had sufficient replication capacity for being used in our in vitro assays to study the impact of defined point mutations associated with CTL selection pressure or drug resistance. This allowed us to assess the associated impact on replication capacity for each mutation or combination of mutations in the context of a different viral backbone that potentially better reflects the viral quasispecies on a population level.

We first analyzed the differential impact of a well-defined panel of mutations associated with CTL escape in the HLA-B27-restricted epitope KK10 in *gag* on the consensus *gag* backbone. Introduction of the primary escape mutation R_264_K into the consensus *gag* backbone led to dramatically reduced replication capacity (Figure 3), confirming the negative impact of R_264_K on viral replication that had also been seen in NL4-3 [16]. However, insertion of the entire set of mutations observed in PLHIV with a CTL response targeting KK10 (SARKLM: R_264_K, L_268_M, and compensatory mutation S_173_A outside the epitope) not only led to partial restoration of R_264_K-associated diminished replication as observed in NL4-3 [16] but resulted in infection rates that significantly exceeded that of unmodified consensus *gag*. The introduction of the S_173_A mutation alone did not relevantly impact replication capacity. In conclusion, this suggests a defined sequential development of mutations in vivo, with an initial random S_173_A mutation constituting the prerequisite for selection of R_264_K, i.e., the successful viral escape from CTL response against KK10.

The viral quasispecies within an HIV-infected individual can be imagined as a state of continuous occurrences of random mutations. Yet, the consensus viral variant utilized in our experiments constitutes an approximation of sequence variations most likely occurring more frequently in PLHIV. Given that our consensus *gag* viral variant with SARKLM exceeded replication capacity of the consensus gag virus in the absence of SARKLM, we would postulate that, at times, the overall viral fitness might be optimized in vivo by random mutational events that incidentally facilitate effective escape from specific immune surveillance. This happens not only by evading the CTL response itself but also by improving replication capacity beyond that of average variants within the viral quasispecies.

Having observed a differential impact of mutations associated with CTL escape on the replication capacity of viral variants harboring our *gag* consensus backbone or the NL4-3 backbone, we assessed whether such differences also might exist for drug resistance mutations. We therefore constructed a panel of 10 viral variants with clinically important drug resistance mutations, either in the context of the NL4-3 backbone or the *gag/pol* consensus backbone, and compared their respective replication capacities.

In line with previous results [26,79,98,99,100,101], eight mutations similarly influenced replication capacity regardless of the viral backbone, with L_100_I increasing replication and D_30_N, L_76_V, I_84_V, K_101_P, Q_151_M, N_155_H, and Y_181_C decreasing replication to different degrees. Surprisingly, two mutations had a differential impact on replication depending on the backbone. Mutations K_65_R and M_184_V on the NL4-3 backbone reduced replication capacity by a factor of 0.79 and 0.70, respectively, while increasing it by a factor of 1.37 and 1.40 on the consensus *gag/pol* backbone (Figure 4 and Figure 5).

Mutations K_65_R and M_184_V are commonly detected in PLHIV on treatment with NRTI [75,102,103]. Both mutations have been identified as major drug resistance mutations, with K_65_R conferring high-level resistance to abacavir and tenofovir [104] and M_184_V to both lamivudine and emtricitabine. For both mutations, a negative effect on replication capacity and viral fitness of HIV and SIV, in vitro as well as in vivo, has been described [97,105,106,107,108,109,110,111]. This effect is additive when K_65_R and M_184_V are combined [112].

In contrast to these previous reports, our data show enhanced replication of viral variants that carry the K_65_R or M_184_V mutation in the context of the consensus sequence. This suggests that the viral backbone itself can significantly impact the effect of drug resistance mutations on viral fitness. Certain drug resistance mutations might not only confer a replication advantage due to diminished drug efficacy but also by being associated with optimized replicative fitness per se in specific viral variants. This would enable the viral variant to become dominant within the quasispecies. Therefore, similarly to our observations for the CTL escape mutations, we would postulate that one or more amino acid differences between the consensus sequence and NL4-3 compensate for the potential defect in replication capacity caused by the drug resistance mutations K_65_R and M_184_V.

Since the consensus sequences for *gag* and/or *pol* have been derived by comparing a multitude of primary sequences from PLHIV, it is likely that at least some of the 24 amino acid differences to NL4-3 are also found in circulating quasispecies, potentially in combinations. We would therefore postulate that, in vivo, the emergence of viral variants with mutations to escape selection pressures by antiviral drugs and immune responses is facilitated by commonly existing specific amino acids at certain positions within the viral proteins.

Our study has several limitations. In particular, the fact that the viral constructs with consensus sequences are artificial and unlikely to occur as such among natural quasispecies in vivo potentially restricts the relevance of our observations. This might also be reflected by the observation that the consensus variants displayed an overall reduced replicative capacity compared with NL4-3. However, the aim of our study was to test the influence of specific mutations associated with defined selection pressures within the host in the context of different viral backbones. Thus, our data should be seen as a proof of concept for a complicated interplay between specific mutations and the viral backbone itself both influencing the overall virus sequence. Whether, for instance, the *gag* mutations of the consensus variants alter capsid structure and hamper processes in capsid formation, viral maturation, or viral uncoating remains to be elucidated in future studies. Secondly, we have only analyzed the replication capacity of variants with escape mutations in vitro in the absence of the corresponding selection pressure, for instance, by antiviral drugs. Since viral fitness in the physiological setting is the result of complex divergent and variable forces such as replication capacity, immune responses, host restriction factors, and antiviral treatments, our results cannot be used to exactly predict the propensity of specific variants to dominate or to perish among a viral quasispecies in vivo.

Furthermore, we cannot state if and to what extent the consensus mutations in *gag* and *pol* influence capsid conformation and protein structure, thereby potentially affecting infection rates. This possible effect might be of certain interest, especially for the KK11 epitope-associated immune escape mutations, as they are located in close proximity to one another on the same planar surface of the folded p24 molecule [16].

In summary, we constructed artificial viral variants based on *gag* and/or *pol* sequences representing the average alignment of a multitude of primary sequences from several HIV-1 clade B isolates from PLHIV. We show that specific viral mutations can yield strikingly different results in in vitro replication assays depending on the viral backbones. According to our results, if a point mutation has only been studied phenotypically on a specific strain in vitro, one should be reluctant to reach a conclusion regarding the general effect of this mutation on replication capacity in general. Thus, studying the impact of mutations on viral replication in alternative viral backbones might result in a better understanding of the complex and sometimes divergent interplay between immune- and drug-related selection pressures and virus fitness. Consensus constructs, as introduced here, could be useful in this context, as they are distinct from arbitrary prototype sequences shaped by various selection forces in a specific host.

## Figures and Tables

**Figure 1 viruses-17-00842-f001:**
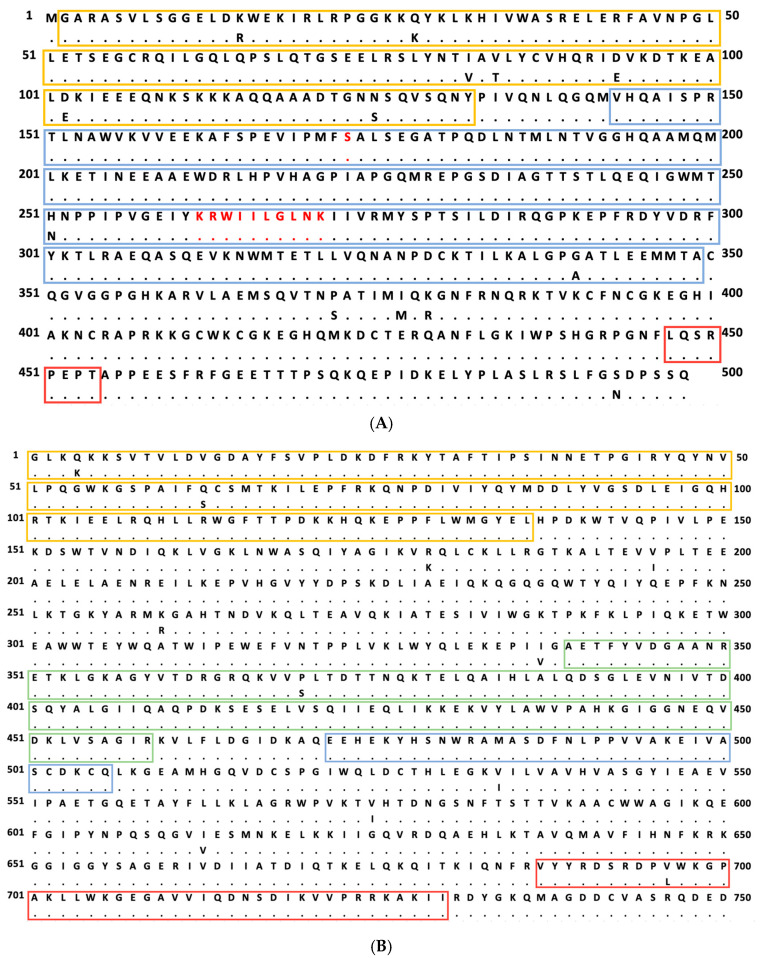
Alignment of amino acid sequences for Gag (**A**) and Pol (**B**) of NL4-3 with the clade B consensus sequence. The upper row represents the NL4-3 sequence. Differences to NL4-3 in the consensus sequence (lower row) are depicted by the one letter code, and identical amino acids are represented by a dot. NL4-3 and consensus sequences differ at 13 amino acid positions in Gag (K_15_R, Q_28_K, I_72_V, V_74_T, D_93_E, D_102_E, N_125_S, H_251_N, G_340_A, P_372_S, I_377_M, K_379_R, S_493_N). The HLA-B27-restricted epitope KK10 (amino acids 263 to 272) and the position of the compensatory mutation S_173_A, associated with CTL escape in KK10, are marked with red letters. In Figure 1A, the boxes mark the different proteins contained. p17 is marked by a yellow box (positions 2 to 132). The section marked by a blue box (positions 143 to 349) comprises the p24 antigen, comprising the N-terminal domain and the C-terminal domain. p6 within Gag is located at positions 447 to 484 (red box). NL4-3 and consensus sequences differ at 11 amino acid positions in Pol (Q_4_K, Q_63_S, R_179_K, V_195_I, K_260_R, I_337_V, P_370_S, V_534_I, V_573_I, I_613_V, V_696_L). The different sections for reverse transcriptase (yellow box, positions 1 to 136), RNase (green box, positions 339 to 459), integrase (blue box, positions 472 to 506), and the integrase-binding domain (red box, positions 687 to 730) are marked, respectively.

**Figure 2 viruses-17-00842-f002:**
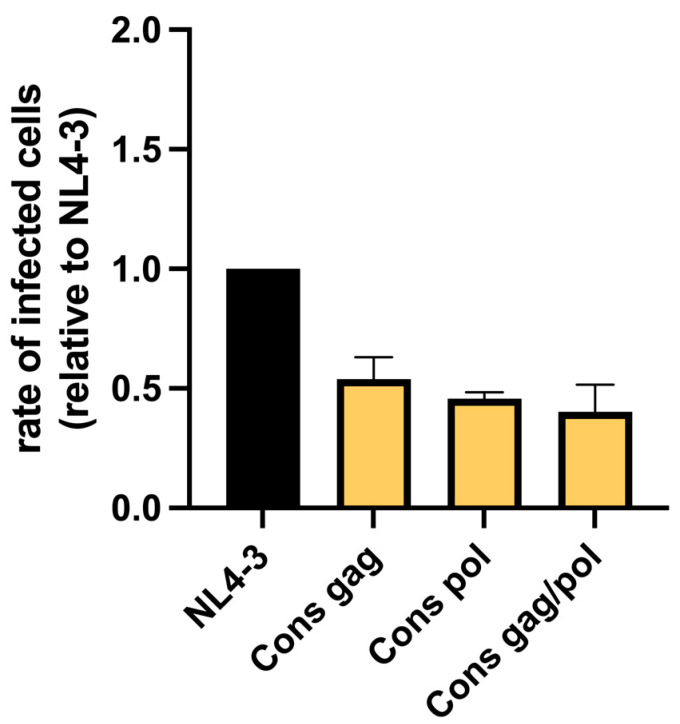
HIV variants carrying consensus sequences of gag and/or pol are replication-competent. The virus backbone in Gag and Pol substantially impacts the effect on in vitro replication capacity of cytotoxic T-lymphocyte-escape mutations and of drug resistance mutations. The three consensus (“Cons”) viral variants replicate at lower levels compared to NL4-3. Cons gag shows an infection rate of 0.54, Cons pol of 0.46, and Cons gag/pol of 0.40 compared to NL4-3, representing significant reductions (*p* < 0.001) for all three clones. Results shown are representative for 3 independent experiments.

**Figure 3 viruses-17-00842-f003:**
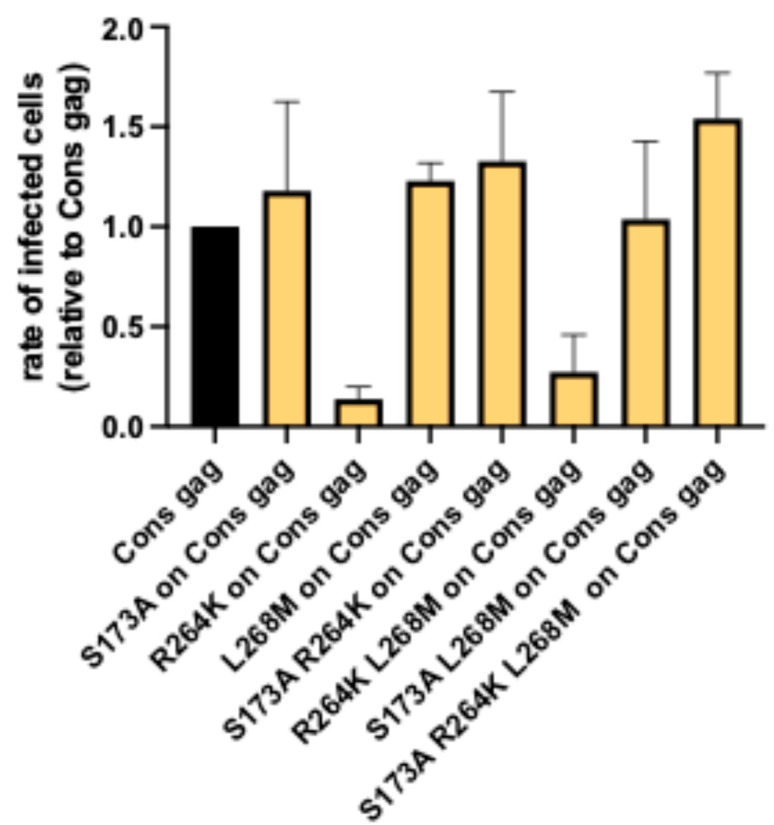
CTL escape mutations on Cons gag viral backbone alter viral replication. CTL escape mutations R264K and L268M and compensatory mutation S173A associated with CTL escape from the selective pressure against epitope KK10 differentially impact infection rates in consensus viruses. A significant impairment in infection rate was observed for the variants carrying the R_264_K mutation (*p* < 0.001) or the R_264_K and L_268_M mutations (*p* < 0.001). Introduction of S_173_A in addition to the primary escape mutations on the Cons *gag* viral backbone enables the variant to exceed replication capacity of the Cons *gag* variant itself. With an infection rate of 1.54 times that of Cons gag in the three experiments carried out, this variant replicated significantly more rapidly than Cons gag (*p* < 0.05).

**Figure 4 viruses-17-00842-f004:**
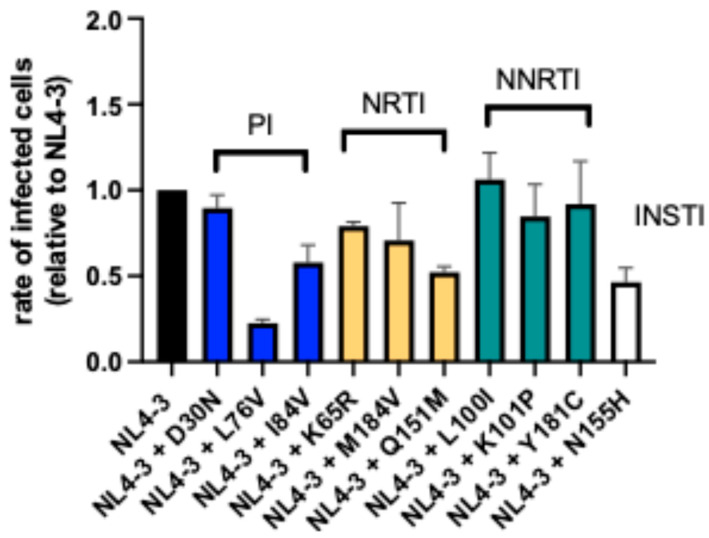
Drug resistance mutations on NL4-3 viral backbone influence infectivity. To analyze the effect of drug resistance mutations on infection rates, a panel of 10 mutations was introduced on the NL4-3 viral backbone. Variants are labeled NL4-3 followed by the respective drug resistance mutation being inserted. For 9 mutations, an impairment in infectivity compared to NL4-3 could be observed, ranging from 0.91 (NL4-3 + Y_181_C) to 0.23 for NL4-3 + L_76_V. Except for NL4-3 + D_30_N and NL4-3 + Y_181_C, the negative effect on infection rates was significant for the remaining variants. The NNRTI-associated drug resistance mutation L_100_I was the only mutation that demonstrated an increase in infection rates (1.06-fold of NL4-3), though it was not significant. The drug resistance mutations introduced are grouped according to the respective drug class. PI: Protease inhibitor; NRTI: nucleoside reverse-transcriptase inhibitor; NNRTI: non-nucleoside reverse-transcriptase inhibitor; INSTI: integrase strand-transfer inhibitor.

**Figure 5 viruses-17-00842-f005:**
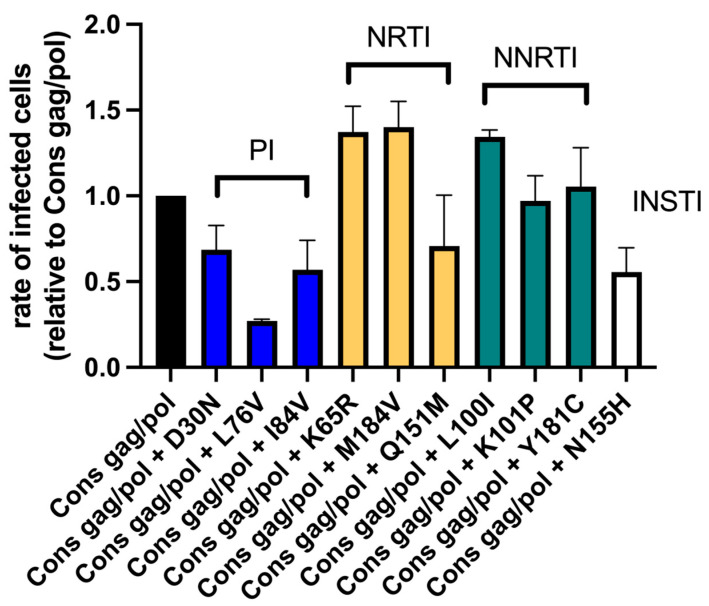
K_65_R and M_184_V drug resistance mutations lead to an increase in infection rates on consensus backbone but not on the NL4-3 backbone. Variants carrying the different drug resistance mutations are labeled “Cons” followed by the respective mutation. Infection rates of the viral variants with drug resistance mutations ranged between 0.27 (Cons + L_76_V) and 1.40 (Cons + M_184_V) compared with the viral variant with clade B consensus sequences in *gag* and *pol* (Cons gag/pol). All drug resistance mutations facilitated a significant change in infection rates, except for mutations K_101_P and Y_181_C. Drug resistance mutations K_65_R and M_184_V on the clade B consensus *gag/pol* variant inversely affected infection rates with a factor of 1.32 and 1.46 as compared with the effect of these mutations on NL4-3 (factor of 0.79 and 0.58, respectively; see Figure 4). The results were grouped according to the respective drug class.

## Data Availability

Data are contained within this article and Appendix A.

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
