# Peer review of "Consensus Sequences for Gag and Pol Introduced into HIV-1 Clade B Laboratory Strains Differentially Influence the Impact of Point Mutations Associated with Immune Escape and with Drug Resistance on Viral Replicative Capacity"

_viruses, 2025, doi:10.3390/v17060842_

Round 1

Reviewer 1 Report (Previous Reviewer 1)

Comments and Suggestions for Authors

This manuscript describes an experimental work showing the importance of sequence context, when assessing the in vitro impact of point mutations within the Gag and Pol polyproteins, from HIV-1 clinical isolates, on the replicative capacity. The study focus on the amino acid sequence context of gag-pol polyprotein. Three viral clones, carrying gag-pol consensus sequences of B-clade viruses were constructed. Using these contexts, ten drug resistant (RT, PR and IN inhibitors) and three immune scape mutations were tested in vitro for their impact on the replication capacity of virus variants.

The problem raised in this study is interesting for the diagnosis of HIV infection, prescription and follow-up treatment. Nevertheless, the study focus on the amino acid sequence context of gag-pol polyprotein though immune scape problem was discussed. However, in order to clarify the significance and implications of the findings from this study, authors should make some important improvements in the manuscript, providing missing information and revising text accuracy.

The conclusion in the Abstract (lines 29-32) overestimates findings of the study, since the immune evasion and drug sensitivity of constructs were not assayed in this work. Associated point mutations within a laboratory adapted or a clinical viral consensus genome sequences might also differ in their impact depending on each sequence context. Indeed, it is necessary to rewrite this sentence more accurately, in order to adjust the conclusion to given data.

Methods: The procedures followed for this study are poorly reported. Authors should revise and further detail this section, improving the reproducibility of the study:

1.       It is not clear how NL4-3 variants, with consensus sequences, were constructed. Lines (107-116): This subsection does not provided enough details. It only mention mutations in p24, while these constructs also carry other mutations in gag and pol.

2.       Its not detailed how drug resistant and immune scape mutations were introduced within the sequence of the three consensus clones.

3.       Lines 140-148: Please, revise thoroughly this method.

            3.1 Authors should explain how GFP-labeled cells, detected by flow cytometry, probe infected cells.

            3.2 The use of this fluorochrome was not mentioned anywhere.

            3.3 Percentages of GFP+ cells were normalized to “wt virus NL4-3”. Also in figures 3 and 5?

Title of figure 4: Please correct –für-

Figure legend 5: (Lines 273-278): Title should be rewritten.

Comments on the Quality of English Language

English is not my first language, I reject to review this aspect.

Author Response

Reviewer 1:

This manuscript describes an experimental work showing the importance of sequence context, when assessing the in vitro impact of point mutations within the Gag and Pol polyproteins, from HIV-1 clinical isolates, on the replicative capacity. The study focus on the amino acid sequence context of gag-pol polyprotein. Three viral clones, carrying gag-pol consensus sequences of B-clade viruses were constructed. Using these contexts, ten drug resistant (RT, PR and IN inhibitors) and three immune scape mutations were tested in vitro for their impact on the replication capacity of virus variants.

The problem raised in this study is interesting for the diagnosis of HIV infection, prescription and follow-up treatment. Nevertheless, the study focus on the amino acid sequence context of gag-pol polyprotein though immune scape problem was discussed. However, in order to clarify the significance and implications of the findings from this study, authors should make some important improvements in the manuscript, providing missing information and revising text accuracy.

The conclusion in the Abstract (lines 29-32) overestimates findings of the study, since the immune evasion and drug sensitivity of constructs were not assayed in this work. Associated point mutations within a laboratory adapted or a clinical viral consensus genome sequences might also differ in their impact depending on each sequence context. Indeed, it is necessary to rewrite this sentence more accurately, in order to adjust the conclusion to given data.

Methods: The procedures followed for this study are poorly reported. Authors should revise and further detail this section, improving the reproducibility of the study: 

  1. It is not clear how NL4-3 variants, with consensus sequences, were constructed. Lines (107-116): This subsection does not provided enough details. It only mention mutations in p24, while these constructs also carry other mutations in gag and pol. 
  2. Its not detailed how drug resistant and immune scape mutations were introduced within the sequence of the three consensus clones. 
  3. Lines 140-148: Please, revise thoroughly this method. 

            3.1 Authors should explain how GFP-labeled cells, detected by flow cytometry, probe infected cells. 

            3.2 The use of this fluorochrome was not mentioned anywhere. 

            3.3 Percentages of GFP+ cells were normalized to “wt virus NL4-3”. Also in figures 3 and 5? 

Title of figure 4: Please correct –für-

Figure legend 5: (Lines 273-278): Title should be rewritten.

Reply:

We greatly appreciate the reviewer’s thoughtful remarks which allowed us to significantly improve data presentation as well as setting up a more readable version.

Therefore, we have revised the linguistic formulations of the manuscript again to make the content more understandable and comprehensible. 

As rightly suggested, we have adjusted title and legend of figure 4 so that the presented data are summarized in a concise manner.  The labeling of the Y-axis has been corrected as well.

In addition, the methods section has been revised and the suggested improvements incorporated. We hope that this will help to explain how the virus variants were generated and how the specific mutations were introduced into the existing sequences. Furthermore, the use of GFP-labeled cells and how they work in this context is now described in more detail.

Reviewer 2:

Comments and Suggestions for Authors

Breitschwerdt et al submitted a revised manuscript that measures the infection capacity of 3 different variants gag, pol, and gagpol that were constructed based on the consensus sequence alignment of HIV-1 subtype B. These variants were replication competent, albeit at much lower capacity when compared to pNL4.3. The authors report a compelling piece of evidence when the immune escape mutations are added in combination with the consensus gag mutations which replicate with improved capacities. The authors also present viral replication data depicting the impact of the drug-resistant mutations from multiple drug classes which revealed mixed results.

Some issues to consider:

Figs 4 and 5 please categorize the mutations with respect to drug class, not numerical order, add brackets to identify this class so the reader can follow the bars easier, the different shades of the bars are not described or what they mean.

Figs 4 and 5, not only categorize these mutants based on drug class, if may be easier to follow and more relevant if these are combined and the differences with or without the Consensus B mutations are combined are shown side by side.

Is this a modified single round assay, because only one round of infection will likely occur, is this truly a replication competent vector? Could you please describe this assay and the purpose of this experiment, wouldn’t it better to do a 30–60-day replication kinetic analyses to determine the fitness of these variants.

Reply:

We deeply appreciate the reviewer’s thoughtful comments and suggestions, which we think have enabled us to significantly improve the manuscript.

We adjusted the figures and sorted the bars according to the associated drug class and marked them accordingly. In addition, we standardized the color of the corresponding bars to achieve a better distinction between the drug resistance classes.

Indeed, in a first version we had combined Figures 4 and 5 and compared the infectious rates of the individual drug resistance mutations on the respective viral backbones side by side. Since this became too confusing due to the larger number of mutations, we had split it up for the respective viral backbone. However, since the reviewer had made a very interesting and extremely helpful suggestion here, we added another table to the supplementaries (Supplementary Table 2), which compares the infectious rates of the individual drug resistance mutations side by side.

In fact, the experiments involve infectious particles, i.e. functional viruses that were previously produced through transfection. From the viral stocks produced by this, a defined amount of viral material (250 ng p24) was used to infect one million of CEM-GXR25 cells. After 48 hours, which equals a single-cycle, the infectivity was determined by the rate of infected cells through FACS analysis.

The suggestion to continue the infection experiment for a longer period and thus to analyze the infection rates and dynamics over a longer period seems particularly interesting with regard to the investigation of viral fitness overall. Unfortunately, we are not able to provide this information but are grateful for this suggestion and will include it in future experiments. In our study the primary goal was, to determine the infectivity of the different viral variants in order to quantify the influence of the viral backbone itself.

Reviewer 2 Report (Previous Reviewer 2)

Comments and Suggestions for Authors

Breitschwerdt et al submitted a revised manuscript that measures the infection capacity of 3 different variants gag, pol, and gagpol that were constructed based on the consensus sequence alignment of HIV-1 subtype B. These variants were replication competent, albeit at much lower capacity when compared to pNL4.3. The authors report a compelling piece of evidence when the immune escape mutations are added in combination with the consensus gag mutations which replicate with improved capacities. The authors also present viral replication data depicting the impact of the drug-resistant mutations from multiple drug classes which revealed mixed results.

Some issues to consider:

Figs 4 and 5 please categorize the mutations with respect to drug class, not numerical order, add brackets to identify this class so the reader can follow the bars easier, the different shades of the bars are not described or what they mean.

Figs 4 and 5, not only categorize these mutants based on drug class, if may be easier to follow and more relevant if these are combined and the differences with or without the Consensus B mutations are combined are shown side by side.

Is this a modified single round assay, because only one round of infection will likely occur, is this truly a replication competent vector? Could you please describe this assay and the purpose of this experiment, wouldn’t it better to do a 30–60-day replication kinetic analyses to determine the fitness of these variants.

Author Response

Reviewer 1:

 This manuscript describes an experimental work showing the importance of sequence context, when assessing the in vitro impact of point mutations within the Gag and Pol polyproteins, from HIV-1 clinical isolates, on the replicative capacity. The study focus on the amino acid sequence context of gag-pol polyprotein. Three viral clones, carrying gag-pol consensus sequences of B-clade viruses were constructed. Using these contexts, ten drug resistant (RT, PR and IN inhibitors) and three immune scape mutations were tested in vitro for their impact on the replication capacity of virus variants.

The problem raised in this study is interesting for the diagnosis of HIV infection, prescription and follow-up treatment. Nevertheless, the study focus on the amino acid sequence context of gag-pol polyprotein though immune scape problem was discussed. However, in order to clarify the significance and implications of the findings from this study, authors should make some important improvements in the manuscript, providing missing information and revising text accuracy.

The conclusion in the Abstract (lines 29-32) overestimates findings of the study, since the immune evasion and drug sensitivity of constructs were not assayed in this work. Associated point mutations within a laboratory adapted or a clinical viral consensus genome sequences might also differ in their impact depending on each sequence context. Indeed, it is necessary to rewrite this sentence more accurately, in order to adjust the conclusion to given data.

Methods: The procedures followed for this study are poorly reported. Authors should revise and further detail this section, improving the reproducibility of the study: 

  1. It is not clear how NL4-3 variants, with consensus sequences, were constructed. Lines (107-116): This subsection does not provided enough details. It only mention mutations in p24, while these constructs also carry other mutations in gag and pol. 
  2. Its not detailed how drug resistant and immune scape mutations were introduced within the sequence of the three consensus clones. 
  3. Lines 140-148: Please, revise thoroughly this method. 

            3.1 Authors should explain how GFP-labeled cells, detected by flow cytometry, probe infected cells. 

            3.2 The use of this fluorochrome was not mentioned anywhere. 

            3.3 Percentages of GFP+ cells were normalized to “wt virus NL4-3”. Also in figures 3 and 5? 

Title of figure 4: Please correct –für-

Figure legend 5: (Lines 273-278): Title should be rewritten.

Reply:

We greatly appreciate the reviewer’s thoughtful remarks which allowed us to significantly improve data presentation as well as setting up a more readable version.

Therefore, we have revised the linguistic formulations of the manuscript again to make the content more understandable and comprehensible. 

As rightly suggested, we have adjusted title and legend of figure 4 so that the presented data are summarized in a concise manner.  The labeling of the Y-axis has been corrected as well.

In addition, the methods section has been revised and the suggested improvements incorporated. We hope that this will help to explain how the virus variants were generated and how the specific mutations were introduced into the existing sequences. Furthermore, the use of GFP-labeled cells and how they work in this context is now described in more detail.

Reviewer 2:

Comments and Suggestions for Authors

Breitschwerdt et al submitted a revised manuscript that measures the infection capacity of 3 different variants gag, pol, and gagpol that were constructed based on the consensus sequence alignment of HIV-1 subtype B. These variants were replication competent, albeit at much lower capacity when compared to pNL4.3. The authors report a compelling piece of evidence when the immune escape mutations are added in combination with the consensus gag mutations which replicate with improved capacities. The authors also present viral replication data depicting the impact of the drug-resistant mutations from multiple drug classes which revealed mixed results.

Some issues to consider:

Figs 4 and 5 please categorize the mutations with respect to drug class, not numerical order, add brackets to identify this class so the reader can follow the bars easier, the different shades of the bars are not described or what they mean.

Figs 4 and 5, not only categorize these mutants based on drug class, if may be easier to follow and more relevant if these are combined and the differences with or without the Consensus B mutations are combined are shown side by side.

Is this a modified single round assay, because only one round of infection will likely occur, is this truly a replication competent vector? Could you please describe this assay and the purpose of this experiment, wouldn’t it better to do a 30–60-day replication kinetic analyses to determine the fitness of these variants.

Reply:

We deeply appreciate the reviewer’s thoughtful comments and suggestions, which we think have enabled us to significantly improve the manuscript.

We adjusted the figures and sorted the bars according to the associated drug class and marked them accordingly. In addition, we standardized the color of the corresponding bars to achieve a better distinction between the drug resistance classes.

Indeed, in a first version we had combined Figures 4 and 5 and compared the infectious rates of the individual drug resistance mutations on the respective viral backbones side by side. Since this became too confusing due to the larger number of mutations, we had split it up for the respective viral backbone. However, since the reviewer had made a very interesting and extremely helpful suggestion here, we added another table to the supplementaries (Supplementary Table 2), which compares the infectious rates of the individual drug resistance mutations side by side.

In fact, the experiments involve infectious particles, i.e. functional viruses that were previously produced through transfection. From the viral stocks produced by this, a defined amount of viral material (250 ng p24) was used to infect one million of CEM-GXR25 cells. After 48 hours, which equals a single-cycle, the infectivity was determined by the rate of infected cells through FACS analysis.

The suggestion to continue the infection experiment for a longer period and thus to analyze the infection rates and dynamics over a longer period seems particularly interesting with regard to the investigation of viral fitness overall. Unfortunately, we are not able to provide this information but are grateful for this suggestion and will include it in future experiments. In our study the primary goal was, to determine the infectivity of the different viral variants in order to quantify the influence of the viral backbone itself.

Reviewer 3 Report (New Reviewer)

Comments and Suggestions for Authors

In this manuscript, authors show that in HIV-1 viral fitness can be affected by the sequence background even in the context of isolates considered as wild-type.  Authors underline the limitation of using a specific strain (NL4-3) in drug resistance studies, suggesting that the presence of specific polymorphisms, including changes affecting CTL escape, can impact the outcome of resistant virus selection. 

The ideas presented in this paper are not new and have been recognized in many labs. The main problem of these studies is the quantification of the contribution of different polymorphisms in resistance emergence and viral fitness in the context of an infection.

There are a number of issues that in my opinion have to be improved in a revised version of the manuscript:

  • The abstract needs improvement. The word consensus is repeated up to 5 times in lines 20-27.
  • Introduction, line 81: An isolate is in fact a quasispecies. I guess authors are probably referring to a clone.
  • Methods, section 2.2: It is not clear to me how the mutant viruses were constructed. If I understood correctly, authors have used NL4-3 as template, and they have introduced 13 mutations in gag and 11 in pol.  Obviously they were not obtained in a single mutagenesis experiment, and different intermediates had to be prepared. The process is not clearly explained (maybe it is complex and should be part of Supplementary materials). 
  • Although authors provide statistical data in the supplementary information, relevant data should be indicated in the legends to figures. For example, p values for the comparison of Cons gag, Cons pol and Cons gag/pol vs. NL4-3 should be provided in Figure 2, etc.
  • The notation of the mutants in Figure 3 is confusing SA should be S173A, while RK should be R264K and LM is L268M. What happens if the opposite mutations are introduced in NL4-3? This experiment would be important to provide significance to their findings. Are these mutations located in the capsid protein CA?  Is there any explanation based on available structures of the proteins?
  • K65R, M184V and Q151M have deleterious effects on fitness according to literature (Bleiber et al. J Virol 2001; 75: 3291-3300; Cong et al. J Virol 2007; 81: 3037-3041; Paredes et al. J Virol 2009; 83: 2038-2043; Wagner et al. Sci Rep 2012; 2: 320). Some discussion and critical analysis is required here,
  • The same applies to mutations associated with resistance to PR inhibitors, particularly D30N and L76V (Dykes et al. J Clin Microbiol 2010; 48: 4035-4043, and others).
  • Typo at line 339: “replication competent”
  • Considering that these are in vitro experiments, it is difficult to extrapolate how the obtained results would be affected in the context of CTL escape mutations.
  • I think that Supplementary information should be in a different file, not in the main body of text.

Author Response

Answer to the reviewer:

We greatly appreciate the reviewer’s thoughtful remarks and have attempted to address all your specific criticisms. We believe that these modifications have significantly improved data presentation and have resulted in a more concise and readable discussion.

As suggested, we have made linguistic adjustments to the abstract and introduction. The description on how to construction the mutant viruses within the methods section has also been revised, and is in more detail now. This section in particular should contribute to a better understanding overall. In addition, we have introduced significance values for the figure captions and we have adapted the labels in Figure 3. In addition, the supplementary section has been added to a different file.

Concerning the KK10-associated immune escape mutations, this panel of mutations and their influence on replication rates had been investigated before by Schneidewind et al.. In our Discussion section, lines 361-384, we are referring to that and are classifying our results in the context of the existing data.

All 3 of the KK10-associated immune escape mutations are located within the N-terminal domain of the p24 antigen. The coding regions of the different proteins were marked in figure 1A by different colored boxes. Now, we have adapted the figure 1A, so that the p24 antigen encoding region (comprising of the C-terminal and N-termninal regions) is marked by a blue box. The extent to which these mutations influence protein confirmation in the context of the consensus virus cannot be determined by our experiments. In order to address this topic, we have added a section to the discussion (lines 438-442).

As stated in the discussion section, lines 421-437, one limitation of our experiments is, that they are in vitro experiments with artificial consensus sequences. Accordingly, we can only make statements about the replication rates of these virus mutants that allow limited conclusions to be drawn about their behavior in vivo. The behavior of CTL escape mutations has already been extensively studied in previous investigations (Schneidewind et al.). Although the viral backbone did show an influence on replication rates compared to the NL4-3 backbone in our experiments, the effect of the mutations themselves appears to continue to be decisive for infection rates.

We are grateful, that the reviewer has emphasized the importance of the negative influence of drug resistance mutations on viral replication. In our manuscript (lines 391-413), we had already discussed this topic, but now tried to be more precise and concise in our statement. The references have been incorporated into this section of the discussion, although the paper of Cong et al. had already been included in the references in the previous draft.

Round 2

Reviewer 3 Report (New Reviewer)

Comments and Suggestions for Authors

Authors have satisfactorily responded to my previous queries and criticisms and the manuscript has been improved.

I propose two minor corrections:

(1) the abstract is cut between lines 23 and 24, and the text should be placed in a single paragraph.

(2) Table S2 in the supplementary section should not be divided into two pages, but placed in a single page.  

Author Response

We are very thankful for the reviewer's positive feedback and appreciate the helpful comments. The suggestions have been implemented and the manuscript has been adapted accordingly. In addition, a caption to Supplementary Figure 3 has been added. 

This manuscript is a resubmission of an earlier submission. The following is a list of the peer review reports and author responses from that submission.

Round 1

Reviewer 1 Report

Comments and Suggestions for Authors

This manuscript describes an experimental work showing the importance of sequence context, when assessing the in vitro impact of point mutations within the Gag and Pol polyproteins, from HIV-1 clinical isolates, on the replicative capacity.

The problem raised in this study is interesting for the diagnosis of HIV infection, prescription and follow-up treatment. However, as it is, the quality of the manuscript is low, in terms of its organization and presentation. Some information is presented in the wrong section. Authors should revise the content of Results and Discussion sections and restructure it, following journal instructions for manuscript preparation. Greater accuracy and text rephrasing will help to make it more understandable and readable.

Specific comments:

1-      Figure 1: 1A and 1B sequences are only part of Gag and Pol precursors. Please, localize the mature proteins (i.e., p17, p24 CA, p6, RT, IN) affected by differences with consensus sequence or by the incorporation of mutations. This information should be mentioned in the main text, Abstract and Results and when discussed their potential impact on the immune scape, drug resistance or viral fitness.

2-      Results section lacks description and comments about figure 3, albeit the discussion section includes some comments about this figure.

3-      It is striking that the legend of figure 2 summarize the data presented, but not in equivalent figures 3-5. A title or a short summary should be added in every figure legend. Apart from that, information about the procedures that all experiments have in common should be moved from the figure legends to the Materials and Methods section.                                                                                                         

4-      Figures 2-5 have a common error: The Y-axis of figures 1-5 says “% infected cells (relative to…)” and units are 0, 0.5, 1, 1.5 and 2. Please, correct units or just remove the percent sign from the legend in order to represent correct values.

5-      Discussion section should be edited in depth. Lack of punctuation marks and very long and complex sentences made difficult to follow the interpretation of results (lines 323-440).

Reviewer 2 Report

Comments and Suggestions for Authors

The authors present a manuscript where a construct a series of constructs that feature mutations at positions  gag and pol representing an average alignment of a primary sequences from several HIV-1 clade B infected individuals. The introduction and writing and presentation of the question at hand is sound and clear. However, there are issues with results and just overall direction of manuscript.

How many primary sequences were aligned?

Interesting to know Frequency of all these mutations occurring at once, or just general frequencies of certain mutations pertaining gag/pol?

Statistical significance of cons gag + specific gag mutations, might not be the different after all due to large standard deviations or error.

Please add titles to figure legends.

Reviewer 3 Report

Comments and Suggestions for Authors

The study employs an HIV-1 consensus gal/pol sequence clone to explore the impact of HIV-1 backbone variability on virus replication following the accumulation of resistance/escape mutations. While this question is crucial in HIV-1 biology, the study faces several significant issues. Firstly, the use of an artificial construct raises concerns about its biological relevance, as acknowledged by the authors in the discussion section. Second, the authors claim contrasting results between virus replication of escape mutants engineered in  NL43 background vs consensus gag/pol background; however, I don’t see a substantial difference and this holds for both CTL escape mutations and drug resistance mutations.   Lastly, the experimental setup appears to be preliminary as conclusions are drawn only from one cell line and one assay.

Comments on the Quality of English Language

The manuscript requires improvement in its English for publication. Throughout the manuscript the authors have used long sentences, making it challenging to comprehend. There is redundancy in the results section in several places. 

Reviewer 4 Report

Comments and Suggestions for Authors

In vivo HIV1 replicates in the form of viral quasispecies that show a high degree of genetic diversity. After infection HIV1 can replicate to very high levels in the lymph nodes fueling point mutations due to the fact that RT  has no proofreading activity. In addition the high level of replication can result in cells undergoing multiple concomitant infections . Such cells can produce heterozygous viruses together with homozygous viruses.

In vivo replication of these homo and heterozygous viruses generates new variants where recombination is probably the major driver.That being said it seems to me that investigating the impact of point mutations on the replicative fitness of mutant viruses (with 1 to 7 point mutations, see materials and methods) in transformed cells ex vivo won’t be of any  help to..understand emergence and  kinetics of viral diversity, see lines 44 vivo and how it impacts viral fitness and virus diversity

Comments on the Quality of English Language

Some english editing is required ,

See for examples 

Lines 221, sentence looks weird

See also lines 415 and 438